# Research on Dual-Technology Fusion Biosensor Chip Based on RNA Virus Medical Detection

**DOI:** 10.3390/mi13091523

**Published:** 2022-09-14

**Authors:** Jin Zhu, Yushan Xie

**Affiliations:** 1Ocean College, Jiangsu University of Science and Technology, Zhenjiang 212003, China; 2College of Electronic Information Engineering, Nanjing University of Aeronautics and Astronautics, Nanjing 211100, China

**Keywords:** quartz crystal microbalance technology, localized surface plasmon resonance, biosensor chip, nanomaterials

## Abstract

In recent years, the emergence of COVID-19 and other epidemics caused by RNA(ribonucleic acid)-type genetic viruses has aroused the close attention of governments around the world on emergency response to public safety and health emergencies. In this paper, an electrodeless biosensing detection chip for RNA virus medical detection is designed using quartz crystal microbalance technology and local surface plasmon resonance technology. The plasmonic resonance characteristic in the nanostructures of gold nanorods-quartz substrates with different parameters and the surface potential distribution of the quartz crystal microbalance sensing chip were studied by COMSOL finite element simulation software. The results show that the arrangement structure and spacing of gold nanorod dimers greatly affect the local surface plasmon resonance of nanorods, which in turn affects the detection results of biomolecules. Moreover, high concentrations of “hot spots” are distributed between both ends and the gap of the gold nanorod dimer, which reflects the strong hybridization of the multiple resonance modes of the nanoparticles. In addition, by simulating and calculating the surface potential distribution of the electrode area and non-electrode area of the biosensor chip, it was found that the biosensor chip with these two areas can enhance the piezoelectric effect of the quartz chip. Under the same simulation conditions, the biochip with a completely electrodeless structure showed a better sensing performance. The sensor chip combining QCM and LSPR can reduce the influence of the metal electrode on the quartz wafer to improve the sensitivity and accuracy of detection. Considering the significant influence of the gold nanorod dimer plasma resonance mode and the significant advantages of the electrodeless biosensor chip, an electrodeless biosensor combining these two technologies is proposed for RNA virus detection and screening, which has potential applications in biomolecular measurement and other related fields.

## 1. Introduction

There are many types of infectious diseases caused by RNA genotype viruses. Most of the diseases are seriously contagious and even cause a global pandemic trend. The outbreak of COVID-19 has become an unpredictable health and safety emergency, reflecting its rapid spread, wide impact, and high mortality. Especially when an outbreak of infectious diseases occurs, the inability to diagnose and quickly judge the attributes of the virus will lead to the further spread of the epidemic, thus missing the golden period of epidemic prevention and control, and causing great panic in society [1]. In recent years, local surface plasmon resonance (LSPR) [2] and quartz crystal microbalance (QCM) [3] technology have been applied to the field of biochip detection, and have attracted the attention of researchers all over the world.

Local surface plasmon resonance is an optical detection technology that can realize label-free, dynamic, and real-time monitoring of the primer probe and the target biomolecule. When light is incident on the gold nanoparticle material composed of noble metal elements, and the vibration frequency of the incident photons matches the overall vibration frequency of the noble metal nanoparticle material or metal conduction electrons, the gold nanoparticles or metal surface will react to these photons. The energy produces a strong absorption effect, which leads to the phenomenon of localized surface plasmon resonance, and a strong resonance absorption peak appears on the spectrum. At present, many methods use LSPR technology for non-invasive detection. In the literature [4], an electrochemiluminescent glucose sensor was prepared using ITO (Indium Tin Oxide) as the electrode base and a chemically synthesized Au/TiO_2_ nanocomplex as the electrode modification material and was successfully applied to the detection of glucose in human serum and saliva, providing a simple low-cost fabrication method for the non-invasive monitoring of disposable glucose. Using the surface effect, volume effect, quantum size effect, and macroscopic quantum tunneling effect of nanomaterials, Li Ying et al. designed an electrochemical biosensor based on nanomaterials modification, which has the performance of a larger current response, higher sensitivity, and lower detection limit to meet the needs of non-invasive blood glucose detection, providing a new detection method for non-invasive blood glucose detection [5]. The author of [6] discusses micro- or nanostructures with localized optical field enhancement, extraordinary light transmission, interference of surface plasmon waves, plasmonic cavities, etc. By summarizing and comparing their performances, guidelines for the design of SPR (surface plasmon resonance) sensors are presented [6]. Furthermore, gold and silver nanospheres serve as starting points for evaluation of the plasmonic optical properties of nanostructured sensor platforms, and related studies found that the tunability of the LSPR spectrum depends on the shape, size, composition, and surrounding dielectric of the nanoparticles [7,8,9]. Theoretical analysis shows that for nanoparticles with a radius of 5 to 40 nm, a large number of LSPR peak shifts will be observed due to the adsorption of the molecular layer on the surface of the nanoparticles. In terms of molecular sensing, the LSPR peak shift is also determined by the thickness of the adsorbed molecular shell. In addition, significant LSPR peaks were observed for gold and silver nanospheres with a radius of 40 nm [10]. Moreover, these anisotropic nanoparticles exhibit a longitudinal LSPR band on top of the LSPR band in the blue range common to spherical nanoparticles, which is superior in all aspects, especially in sensitivity to surrounding media and LSPR biosensing aspects [11,12]. Among the various types of metal nanoparticles, gold nanorods (AuNRs) have drawn particular attention in the development of sensitive LSPR biosensors due to their own unique optical and photothermal properties [13,14,15,16]. As a new class of anisotropic nanomaterials, AuNRs differ greatly from spherical nanoparticles because they show two characteristic transverse and longitudinal surface plasmon absorption bands. This absorbed light can be used as an energy acceptor in fluorescence resonance energy transfer assays, or to transfer light into heat for photothermal therapy of cancer. In particular, the controllable spatial arrangement of AuNRs induces changes in optical properties, making them particularly suitable for sensing various analytes. The application of AuNRs in the fields of cell imaging and cancer therapy has attracted intense interest [17]. Regarding the use of localized surface plasmon resonance technology for virus detection, Funari et al. developed an optical microfluidic sensing platform with gold nanopins prepared by electrodeposition using the sensing mechanism of LSPR technology to detect the amount of antibodies to SARS-CoV-2 burst protein in 1 μL of human plasma diluted in 1 ml of buffer solution for approximately 30 min. The local refractive index change caused by antigen–antibody binding causes a change in the position of the peak nanometer LSPR wavelength [18]. Zeinoddini et al. exploited the LSPR properties of the nanoparticle surface by attaching viral antibodies directly to the surface of the nanoparticles by UV-visible spectroscopy (Perkin Elmer Lambda 20), dynamic light scattering (DLS) detection methods, and direct measurement of zeta potential gradients (Zp), which for the first time verified and revealed processes such as binding between antibody molecules and viruses [19]. Achadu et al. developed a novel magnetic/plasma-assisted fluorescent immunoassay system in which influenza A virus-specific antibodies were first coupled to the surface of nanoparticles in the presence of influenza A virus (as the test virus), and an immune complex formed between the antibody and the nanomaterial. Their interaction resulted in a gradual increase in the fluorescence intensity of the detection probe with increasing influenza virus concentration [20]. In addition, nanoparticle-based SPR sensors have been widely used in various fields such as cancer diagnosis, virus detection, molecular response, and environmental monitoring due to their high sensitivity and specificity [21,22,23,24,25].

QCM has the advantages of high sensitivity, fast response, real-time measurement, good operability, and low production cost. It realizes the qualitative and quantitative detection of the physical properties such as the adsorption amount of the measured biomolecule through the change in the resonance frequency caused by the chemical adsorption reaction between the primer probe and the target biomolecule. Some researchers have successfully achieved the direct detection of salivary glucose by endowing the microgels on QCM chips with excellent protein resistance and glucose sensitivity properties to obtain a simple device for glucose detection, which is also useful for trace amounts of small organic molecules. The detection of glucose provides an important solution with the potential to advance the practical application of QCM sensors [26,27]. In addition, Lim, Ji Yoon, and others have developed a technology for sensitive and selective detection of miR-21 molecules using a quartz crystal microbalance biosensor, and the developed QCM biosensor is very effective for quantification of miR-21 in serum samples at an early stage. The diagnosis of diseases, such as cancer and vascular diseases, offers great potential and can be a good alternative for biological research and clinical diagnosis [28]. The QCM immunosensor based on magnetic nanobead amplification developed by DujuanLi et al. provides a rapid, sensitive, and specific method for the detection of AIV H5N1 in agricultural, food, environmental, and clinical samples [29]. Ronghui Wang et al. developed a quartz crystal microbalance (QCM) passive sensor based on ssDNA cross-linked polymer hydrogels that can be used for rapid, sensitive, and specific detection of avian influenza virus (AIV) H5N1 [30].

In addition, quartz crystal microbalance technology can be used for label-free biosensor detection of trypsin [31], immunosensor detection [32], high-sensitivity gas detection of volatile sulfur compounds [33], and in medical diagnosis, environmental monitoring, food inspection, and other fields. QCM and LSPR are two new technologies for real-time detection of the physical and chemical parameters of biofilm. A collaboration between Stanford University (USA, Stanford, California) and the Max Planck Institute for Polymer Research (Ackermannweg 10, 55128 Mainz, Germany) investigated the use of a combination of surface plasmon resonance (SPR) and a quartz crystal microbalance (QCM) to monitor in situ the solution phase adsorption of the perfluoropolyether lubricant Fomblin ZDOL on silver surfaces [34]. Professor Daming Zhu from University of Science and Technology of China (China, Jiangsu, Anhui), Academician Yao Shouzhu from Hunan Normal University (China, Hunan, Changsha), and Prof. Qingji Xie et al. carried out detailed analyses of the adsorption of probe molecules at the solid–liquid interface using quartz crystal microbalance (QCM) and surface plasmon resonance (SPR) spectroscopy responses, respectively [35,36,37]. A comparison of the analysis of the two techniques yielded experimental results that are difficult to obtain by a single technique. Although they have different detection mechanisms, they have obvious similarities in test objects, detection resolution, and other aspects, and their structures are basically compatible, which can be integrated on the same sensing and detection platform. Although the two new sensing and detection technologies have their own unique advantages in the fields of biological, chemical, medical, and other analysis and detection, they each have obvious shortcomings. LSPR detection technology needs to have significantly different refractive indexes at the interface between the solid [18] and the liquid. The biomolecular mass detected by QCM includes the mass of the coupling solution.

The simple relational expressions used by each testing technique are approximated under certain assumptions, so the calculated results will have large or small deviations from the actual situation. This is realized by the combination of two independent measurement systems, in which the equipment is complex and it is difficult to achieve synchronous measurement, which makes the results unable to achieve the true synergy and complementary effect [34,35,36,37]. The purpose of this study was to study the radiation characteristics of the plasma field of double gold nanorods on the biochip and the surface potential distribution of the biochip, then select the appropriate chip size and structure, and finally combine the two technologies on the same chip. Firstly, the surface extinction spectral characteristics of gold nanorods with different array spacings (d) were preliminarily analyzed for two different gold nanorod dimers (side-by-side arrangement and longitudinal arrangement). Then, the surface potential distribution of the biosensor chip under a polar structure (gold electrodes assembled directly to the quartz wafer surface) and electrodeless structure (gold electrodes peeled off the quartz wafer surface with small spacing) was studied. Through modeling and simulation, the arrangement of gold nanorod dimers on the biosensor chip, and the structure and size parameters of the chip were determined, forming an electrodeless dual sensor technology biochip. The biosensor chip studied in this paper adopts multi-technology synchronous measurement to realize the detection and observation of the target test object from multiple angles at the same time. The obtained information can overcome the deviation caused by the assumption of a single detection technology through complementation and correction, which is more conducive to improvement of the reliability and accuracy of detection information.

## 2. Materials and Methods

With the continuous exploration of the properties of noble metal nanostructures, various noble metal nanostructures have been successfully prepared experimentally. For these complex nanostructures, Maxwell’s equations cannot be used to solve them analytically. In 1908, Gustav Mie proposed a set of electromagnetic wave scattering theory for spherical particles that is still widely used so far [38]. With the increasing development of nanofabrication technology, nanostructures such as rods and sheets have appeared [39,40,41], and the Mie theory, only used for spherical particles, no longer meets people’s needs. Gans [42] performed appropriate corrections and extensions to the Mie theory, so that it can effectively solve the scattering problem of similar ellipsoid particles. The emergence of the Mie and Gans theory makes the scattering problem of conventional spherical and ellipsoidal particles well resolved. However, the above two theories cannot solve the extinction, absorption, and scattering problems of non-spherical and non-ellipsoidal noble metal nanostructures. In order to solve this problem, many practical numerical simulation methods have been developed to solve the above optical properties of noble metal nanostructures, such as: discrete dipole approximation (DDA), time domain finite difference method (FDTD), finite element method (FEM), etc. Effective numerical simulation methods can provide reliable data reference for experiments, and can easily design and optimize noble metal nanostructures, and then explore their potential application value.

In this study, the spectral and electric field distributions of different arrangements of gold nanorod dimers were numerically simulated by COMSOL Multiphysics 5.5 (Stockholm, Sweden). The most basic idea of the finite element method (FEM) was proposed by R.L. Courant in 1954 [43]. The basic solution idea is to hypothetically partition the continuum to be analyzed into a finite ensemble of cells, and then use the interpolation function to determine the field function on the ensemble of cells. If the cells meet the convergence requirements of the problem, then as the size of the cells is reduced and the number of cells in the solution region is increased, the approximation of the solution will continue to improve, and the approximate solution will eventually converge to the exact solution. As shown in Figure 1, because the quartz substrate of the chip destroys the uniformity of the background field of the gold nanorod dimer, a two-step method is used in the finite element calculation to analyze the plasma characteristics of the nanostructure. The first step is to calculate the three-layer dielectric structure using the Fresnel equation, including the environmental dielectric layer (n_a_ = 1), quartz substrate (mainly composed of SiO_2_), and intermediate polyelectrolyte layer (PE, n_b_ = 1.48) [44]. The polyelectrolyte layer is located between the quartz substrate and the gold nanorod dimer. The second step is to set the gold nanorod dimer to the activated state, use the background field obtained in the first step as the excitation source, calculate the scattering and absorption of the gold nanorod dimer under different arrangement structures, and set periodic conditions around the gold nanorod dimer.

As shown in Figure 1, Au (gold) (Johnson and Christy 1972: n, k 0.188–1.937 µm) dimers consist of a cylinder and two hemispheres. The height of the nanorods is 75 nm and the diameter of the hemispheres is 30 nm. In order to reduce reflection interference and improve the convergence speed of calculation, an 80-nm-thick perfectly matched layer (PML) is wrapped around the outermost edge of the physical domain. The air domain has a length of 450 nm, a width of 400 nm, and a height of 200 nm. The substrate material is quartz (the main component is SiO_2_), and the length, width, and height are 400, 400, and 45 nm, respectively. The interstitial layer between the nanorods and the quartz substrate is defined as an intermediate polyelectrolyte layer (PE, nb = 1.48) with a thickness of 1.5 nm [44]. The direction of the incident light is perpendicular to the plane of the gold nanorod dimer. The gold nanorod dimer is surrounded by air. The distance between the dimers of gold nanorods is defined as D = 3 nm.

This paper studies the plasma resonance properties of gold nanorod dimers with different arrangement structures on biochips at different distances D. In the simulation process, only one parameter value is changed each time for the simulation analysis. The research process is as follows: firstly, the extinction spectrum and electric field intensity of gold nanorod dimers arranged side by side with different spacings, D, were studied (Figure 1a); secondly, the extinction spectrum and electric field intensity of gold nanorod dimers longitudinally arranged at different spacings, D, were studied (Figure 1b). The size of the spacings D ranges from 3 to 15 nm in steps of 3 nm. Figure 2 shows the sectional view (Y-Z section) of the two structural models.

The mechanism of QCM for detecting viral nucleic acid molecules is to detect the change in the frequency of piezoelectric materials (such as quartz) caused by specific chemical reactions between target biological molecules through a set of primer probes with recognition and capture functions, and then the sensor converts this change into resonant frequency signal output to realize the qualitative or quantitative detection of target biological molecules through frequency change and calculation of the biofilm’s physical parameters.

In this study, the surface potential distribution of biosensor chips with electrode (Figure 3a) and electrodeless (Figure 3b) structures was numerically simulated. The biosensor chip is composed of a quartz wafer, ring top gold electrode, and circular gold electrode. Because noble metal nanoparticles have stable chemical properties and high optical sensitivity [45,46], the gold nanorod array is placed on the surface of the quartz crystal corresponding to the hollow area of the annular gold electrode. The advantage of using an electrodeless ring electrode is that it makes the assembly of nanoparticles on the quartz wafer easier, and also avoids the corrosion of gold nanoparticles to the electrode by chemical reagents during the assembly process. When the light beam hits the center of the ring electrode vertically, the gold nanorod absorbs light for the first time. Then, the light beam is transmitted through the quartz crystal and irradiates the circular gold electrode at the bottom. The circular gold electrode at the bottom reflects the beam vertically and irradiates it to the gold nanorods again to form the secondary beam absorption of the nanorods to enhance the intensity of the absorption spectrum and improve the optical sensitivity.

The quartz chip in the biosensor chip is cut at (quartz LH (1949 IRE), Renlu Technology Co., Ltd., Shenzhen, China), with an eigenfrequency of 5 MHz, a radius of 12.7 mm, and a thickness of 0.333 mm. The inner diameter of the annular gold electrode is 4 mm, the outer diameter is 8 mm, and the thickness is 100 nm. The circular gold electrode has a radius of 6 mm and a thickness of 100 nm. The reason why the diameter of the top electrode is larger than that of the bottom electrode is that the larger surface area of the top electrode adsorbs more primer probes, and in addition, the breakdown effect caused by the same size of the upper and lower electrodes is avoided.

Because virus samples need to be injected directly into the electrode of biochip, and different biological probes need to be replaced for different viruses’ detection, the chip consumption is increased. Therefore, in order to solve this problem, this paper studied two different biosensor chip structures. First, as shown in Figure 3a, the potential distribution of the electrode biochip was studied. The electrode structure refers to the direct electroplating of the upper and lower gold electrodes on the quartz wafer. Secondly, as shown in Figure 3b, the potential distribution of the upper and lower electrodes of the electrodeless biochip was studied. The electrodeless structure has had the top and bottom electrodes peeled from the surface of the quartz wafer at a very small distance, where the distance D1 between the top electrode and the quart wafer is equal to 2 mm and the distance D2 between the bottom electrode and the quartz wafer is equal to 1 mm.

The QCM chip is used as a biosensor in combination with gold nanorods with LSPR properties for RNA virus detection. The electrodeless structure in this paper is shown in Figure 3b, which means that the gold electrodes on the upper and lower surfaces of the QCM chip in the polar structure are peeled off the QCM chip set on the upper and lower surfaces of the cuvette used for RNA virus detection; this does not mean that no electrodes are present. For RNA virus detection, a primer probe that binds specifically to the target viral nucleic acid molecule is fabricated on the surface of the quartz chip in an electrodeless structure while the electrodes are prepared off the quartz chip on the sample measurement cell without contacting the primer probe, so that only the quartz chip in the chip needs to be cleaned and the primer probe reassembled when performing detection of different RNA viruses. After completing the RNA virus detection, the biofilm multi-parameter calculation data fusion algorithm is used. Through the known quartz wafer density, thickness, and resonant frequency change data, use of the data fusion algorithm can successively obtain the total mass of the adsorbed biofilm, adsorbed biofilm density, adsorbed biofilm viscosity, LSPR electromagnetic field decay length, and other parameters. These parameters are integrated and analyzed and the quality of the coupling solvent can be obtained to improve the accuracy of RNA virus nucleic acid detection.

## 3. Results and Discussion

### 3.1. LSPR Properties of Gold Nanorod Dimers

#### 3.1.1. Parallel Structure

In this experiment, the distance D between the dimers of gold nanorods in the side-by-side arrangement structure in Figure 1b is different, with a variation range of 3–15 nm and a step length of 3 nm. In Figure 4a, there are two absorption bands. The longitudinal resonance peak is in the range of 620~660 nm while the transverse resonance peak is near 340 nm. This trend is similar to that studied by other researchers [47,48].

The long-wave resonance peak shows strongly anisotropic polarization properties associated with the long-axis resonance due to the collective resonance of the free electrons along the long axis enhancing the probability of inter-band radiative excursions, resulting in a greater intensity of their vibrations, while the short-wave resonance does not change significantly. They reflect the effect of the free electrons on the surface of gold nanorods moving along the long and short axes of the nanorods, respectively. Moreover, the hybridization between the localized surface plasmon resonance of the gold nanorod dimers and the propagating surface plasmon of the quartz substrate is very sensitive to the arrangement spacing between the gold nanorod dimers. With the increase in the distance D of the Au nanorod dimers, the resonance peak of the longitudinal mode gradually shifted to red, and the extinction intensity gradually increased. A notable feature in Figure 4a is that when the distance between two nanorods is 3 nm, the extinction intensity decreases sharply. This can be understood as that when the distance between the gold nanorod dimers is too small, the gold nanorod dimer can be equivalent to the gold nanorod monomer with a small aspect ratio. As shown in Table 1, compared with the complex multi polarization mode produced by the gold nanorod dimer, the polarization mode’s complexity and intensity of the gold nanorod dimer with a spacing of 3 nm are reduced.

Figure 4b shows the relationship between the resonant peak position and the distance D of the gold nanorod dimers. The results show that when the distance D increases uniformly, the redshift and intensity of the resonant peak also increase uniformly. The relationship between the resonant peak wavelength (*y*) and the distance D (x) is fitted with a linear function to obtain the following formula:(1)y=626.79+0.93x

In order to better reveal the optical and related electromagnetic properties of gold nanorod dimers, the electric field distribution on the surface of gold nanorods at formants with different spacings, D, was simulated, as shown in Table 1. This paper mainly observes the electric field components Ex, Ey, Ez, and electric field |E| under different distances, D.

**Table 1 micromachines-13-01523-t001:** Electric field distribution under the resonance peak of the gold nanorod dimer in the juxtaposed structure (Y-Z section).

AuNRs_GAP	3 nm	6 nm	9 nm	12 nm	15 nm
Electric FieldResonance Peak	629.4 nm	631.8 nm	634.4 nm	637.6 nm	640 nm
Ex	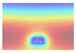	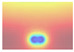	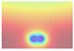	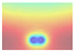	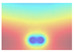
Ey	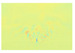	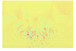	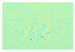	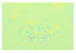	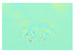
Ez	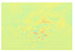	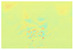	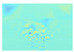	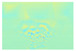	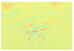
|E|	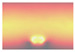	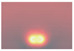	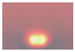	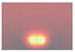	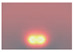

In Table 1, as the distance D between the Au nanorod dimers increases, the electric field component of the Y-Z cross-section changes very little. It can be inferred that the excitation of the transverse mode (the excitation field mode in the short axis direction of nanorods) is almost independent of the distance D. In addition, the electric field component Ex radiated into space by the gold nanorod dimer has an obvious weakening trend, and there are almost no Ey and EZ components in the Y-Z section. In Table 2, it can be seen from the electric field component diagrams of Ex and Ez that the increase in D gradually enhances the radiation field in space. In the electric field mode |E|, the electric field intensity at both ends of the gold nanorods and in the gap shows a slight weakening trend. Due to the anisotropy of the nanorods, different charge distributions are generated on the surface of the nanoparticles. The upper surfaces of the gold nanorods all indicate the appearance of quadrupole modes. In Table 3, the surface of the gold nanorods also shows a quadrupole mode, which is consistent with the pictures of the Ex and Ez components in Table 2. The regions with high brightness in the |E| component diagram are called “hot spots”. From Table 1, it can be found that there are hot spots with higher electric field concentrations between both ends of the gold nanorods and between the gaps. The reason for this electromagnetic focusing effect is that the existence of the quartz substrate destroys the scattering properties of rod-shaped gold nanoparticles, and there is a close connection between gold nanorods and quartz.

#### 3.1.2. Vertical Structure

In this experiment, the plasma resonance characteristics of the vertically arranged nanorod dimers were studied, as shown in Figure 1a. The size range of the spacing D is 3~15 nm, and the step length is 3 nm. Figure 5a shows the plasma optical properties of this structure, which includes two absorption bands: the longitudinal (along the long axis of the nanorod dimer) formant is in the range of 700~760 nm, and the transverse (along the short axis of the nanorod dimer) formant is near 390 nm. With the decrease in the spacing D, the resonant peak of the longitudinal mode redshifts and the amplitude of the resonant peak increases. When the interval D decreases uniformly, the redshift speed of the resonant peak and the reduction rate of the amplitude are significantly accelerated. This phenomenon may be related to the increase in the phase delay between the gold nanorods and the quartz substrate, which makes the resonant peak red shift and the intensity of the resonant peak increase.

The relationship between the resonant peak position (*λ*) of the extinction cross-section of the nanorod surface of the AuNRs-PE-quartz structure in the longitudinal arrangement and the arrangement spacing D (AuNRs_GAP) is shown in Figure 5b. It can be found from the figure that when the interval D between the gold nanorod dimers increases uniformly, the red shift speed and the amplitude increase rate of the resonance peaks increase significantly. Here, a polynomial function is used to fit the dataset with the relationship:(2)$$\lambda =785.04-7.997\;x+0.3156\;x^{2}$$

At the same time, the electric field distribution at the resonance peak of the lower surface of the longitudinally arranged gold nanorod dimer with different arrangement spacings, D, was also simulated, as shown in Table 4, Table 5 and Table 6.

In Table 4, the electric field component of the Y-Z cross-section shows an enhanced trend with the increase in the spacing, D. It is inferred that the excitation of the transverse mode in this arrangement structure depends on the change in the arrangement gap. In the Ex and Ez components in Table 5, the plasma field gradually radiates to the space outside the dimer of gold nanorods, and the electric field components at both ends of gold nanorods and at the gap show an increasing trend. The polarization mode of the quadrupole mode appears on the surface of gold nanorods. In the Ex and Ez component diagram of Table 6, the surface of the gold nanorods also shows the quadrupole mode, but the polarization intensity is weaker than the Ex and Ez components in Table 5, and the Ey component in the X-Z section does not show polarization. In addition, the distribution of regions with high brightness in the |E| component map in Table 2 is consistent with that in Table 1.

The following conclusions can be drawn from the simulation results of the above two different nanorod arrays: the quantum size effect and surface effect of nanoparticles are the main factors leading to the red or blue shift of the extinction cross-section of nanoparticles. The peak shift of the extinction cross-section of the metal nanorod dimer is mainly caused by the surface effect. Since the longitudinal plasmonic modes of the nanorods have a higher oscillation intensity than the transverse modes, the electric field component parallel to the length of the nanorods has the highest scattering efficiency. Additionally, when the gold nanorod dimers are close to each other, the dimers can be regarded as monomers. In addition, the electric field direction can be regarded as the long axis direction of the nanorods. When the direction of the electric field is along the *X* axis, the intensity of the longitudinally arranged Au nanorod dimers is higher than that of the side-by-side Au nanorod dimers. It can be seen from the observation that the juxtaposed gold nanorod dimer can be regarded as the monomer of the nanorod along the direction of the electric field, as the volume increases and the aspect ratio decreases. However, longitudinal gold nanorod dimers, which are monomers of nanorods, increase in volume and aspect ratio. Gold nanorods are highly sensitive to changes in spacing. At smaller spacings, the electric field is mainly confined between the ends of the nanorods and the gap. These regions show enhanced local electric field effects. There are strong dipole resonances along the long axis of the nanorods. The arrangement of the gold nanorod dimer affects the plasmon-mediated optical sensing and enhancement to a certain extent, and the rational arrangement of the nanorod arrangement structure and arrangement spacing can greatly improve the sensitivity of the nanobiosensor.

### 3.2. Research on Biosensor Chips

Figure 6 shows the simulation model (Figure 6a,c) and potential distribution (Figure 6b,d) of the biosensor chip with the ring round gold electrode. Figure 6a shows the simulation model of the biosensor chip with the electrode, and Figure 6c shows the simulation model of the biosensor chip with an electrodeless structure. When 10 V of voltage is applied to the upper and lower gold electrodes of the biosensor chip with electrode structure, it can be seen from Figure 6b,d that most of the high potential is concentrated in the “electrode area” while the potential of the “electrodeless area” is very small, and the potential intensity decays exponentially with the increase in the distance from the center of the electrode. This phenomenon is called the “energy trapping effect”. When the same voltage is applied to the upper and lower electrodes of the biosensor chip without an electrode structure, the electric field generated between the electrodes can still excite the piezoelectric effect of the quartz wafer, and its potential distribution is similar to that of the biosensor chip with an electrode structure. Moreover, the potential in the central region of the annular gold electrode used to assemble gold nanorods is very high, which is more conducive to the improvement of the surface plasmon resonance characteristics of gold nanoparticles.

It can be seen that on the basis of the electrode structure, the use of the electrodeless structure can further reduce the limitation on the resonance of the quartz chip. At the same time, it is easier to make nanoparticles on the quartz chip than on the gold electrode. The cost of replacing the quartz chip is lower, and ultimately improves the stability and service life of the chip. The quartz chip is used as a biosensor for the transmission of information about changes in the resonant frequency of the chip when detecting viruses. The designed primer probes need to be assembled into the middle region of the quartz chip of the sensor before the RNA virus detection is performed. The polaryless structure of the chip needs to be connected to an external drive circuit for RNA virus detection, through which the quartz chip is driven to vibrate at a frequency of 15 MHz. When a sample of the virus to be tested is introduced, if the sample contains a target viral nucleic acid molecule, this target nucleic acid molecule will specifically bind to the nano-priming probe, resulting in an increase in the mass on the surface of the quartz chip. The increase in mass leads to a decrease in the frequency of the quartz wafer according to the Sauerbry formula [49,50]. It was verified through modeling simulations and RNA virus tests that after applying a certain voltage to produce the inverse piezoelectric effect, the high potential remains concentrated in the region of the QCM chip used to assemble the gold nanorods in the electrodeless structure, which provides the energy for the resonance of the QCM chip. Finally, the resonance frequency of the QCM chip with gold nanorods assembled with LSPR characteristics is used to detect the presence of the target virus in the sample solution to be tested. The biosensor chip without an electrode structure is expected to play a certain role in biology, medicine, and other related fields in the future, and has a good development prospect.

### 3.3. RNA Virus Detection Test

#### 3.3.1. Reagents and Instruments

In order to verify the accuracy, sensitivity, and efficiency of the designed RNA virus medical test biosensor chip and test prototype, this paper used infectious RNA viruses for actual testing. The viruses selected for testing were hand, foot, and mouth disease virus samples and norovirus nucleic acid samples, respectively. A total of 4 sets of norovirus (3 positive and 1 negative) and 36 sets of HFMD virus (32 positive and 4 negative) nucleic acid samples were tested. All RNA virus nucleic acid samples for this test were provided by the Zhenjiang CDC (China).

The initial design of the biological probes for RNA virus nucleic acid detection was proposed by Kingsray Biotechnology Ltd (China, Jiangsu, Nanjing). Among the parameters and rules for the design of the probes were: (1) the length of the probes was 20–25 bases; (2) the number of repeats of a single base in each probe should not exceed 6; and (3) the homology of each probe with other viral genomes should not be greater than 50%. After obtaining the probes designed by Kingsley, further screening was carried out to compare the base sequences of the designed probes with those of other RNA-like viruses in order to eliminate probes with greater than 50% homology with other viruses, thus ensuring the accuracy of the test. The final two primer probes used through screening were identified as follows:Double-stranded DNA primer sequence 1 is:G1-SKF: 5′-CTGCCCGAATTYGTAAATGA-3′G1-SKR: 5′-CCAACCCARCCATTRTACA-3′Double-stranded DNA primer sequence 2 is:OL68-1: 5′-GGTAACTTTCCACCACCAATGCCC-3′MD91: 5′- CCTCCGGCCCCTGAATGCGGCTAAT-3′

The medical test instrument used in this experiment was the RNA virus medical test induction biosensor chip and its test prototype designed in this paper. The test was conducted in the virus laboratory of the Municipal CDC, which was equipped with biosafety cabinets and ventilation equipment to prevent the spread of virus aerosols. Protective measures such as wearing a mask and protective clothing were carried out during the tests to ensure the safety of the entire process. The RNA virus medical testing prototype is equipped with a waste liquid collection bin, and the waste water collected at the end of each test was treated under high pressure in the laboratory to ensure the safe disposal of waste water. The experimental platform and environment in which the RNA virus testing was carried out is shown in Figure 7.

#### 3.3.2. RNA Virus Detection Test Method

Primer probes for RNA virus detection were first prepared and probe assembly was carried out by gold-thiol (Au-S) chemical bond coupling to prepare primer probes for DNA-nanogold complexes [51]. In order to shield the charge rejection during the assembly process, the adsorption of the sparse DNA on the surface of AuNRs was achieved by a salt addition aging scheme, i.e., salt (NaC1) was gradually added to the mixture of sparse DNA and AuNRs to induce more thiol DNA, which can be adsorbed on the surface of AuNRs through AuS bonding, which finally completed the preparation of DNA-nanocomposite probes.

The piezoelectric quartz wafer was immobilized with the prepared nanocomposite probes, the control circuitry was switched on, the temperature of the sample was controlled by the temperature control optimization algorithm in the thermostat control system, and the PBS buffer was passed through until the frequency of the induction biosensor chip output from the QCM acquisition system was stable. The frequency at this point was recorded quickly and is the reference frequency f_1_ at no load (before the RNA virus sample is introduced). When ready, the RNA virus nucleic acid sample to be tested was passed into the induction cell and the QCM frequency acquisition system was used to collect the RNA virus nucleic acid sample and the primer probe until the frequency signal from the induction biosensor chip was stable again, at which point the frequency was recorded as f_2_. Based on the two frequency changes (f_1_–f_2_) Hz, the difference between the two frequency changes is the mean value after three measurements for each set of experiments to avoid errors associated with a single measurement. Finally, the biofilm multi-parameter fusion algorithm was combined with the biofilm multi-parameter fusion algorithm to derive multiple parameters of the biofilm, and the properties of the sample to be measured were determined based on the calculated biofilm parameters.

#### 3.3.3. RNA Virus Detection Test Results

In total, 4 groups of norovirus samples (three positive and one negative) and 36 groups of hand, foot, and mouth disease (HFMD) virus samples (32 positive and 4 negative) were tested at 3 different dilution concentrations of 10×, 50×, and 100×, and each group was tested 3 times and then averaged. The final results of the norovirus tests obtained are shown in Figure 8a. The resonant frequency change of the induction bio-sensor chip can be obtained as follows: the resonant frequency of the sensor chip showed a significant decrease compared to the reference frequency f_1_ (approximately 14.9993 MHz) for different dilution concentrations of samples 1–3 in the 4 groups of samples while sample 4 showed no significant change at different dilution concentrations, according to the detection principle of the test equipment design: sample 1–sample 3 is a positive sample and sample 4 is a negative sample. The results of the HFMD virus detection are shown in Figure 8b. Again, based on the detected changes in the resonant frequency of the induction bio-sensing detection chip, it can be obtained that for the 36 groups of samples at different dilution concentrations for sample 6, sample 33, sample 35, and sample 36, no significant decrease in the resonant frequency of the sensing detection chip occurred compared to the reference frequency f_1_ (approximately 14.9993 MHz) Therefore, according to the principle of the test equipment, samples 6, 33, 35, and 36 were negative while the remaining 32 samples were positive. Based on the sample data provided by the CDC, it can be seen that the detection accuracy of the instrument designed in this paper is high.

In addition, according to the test results, the instrument can detect sensitive changes in the resonance frequency for all positive samples at different concentrations: the frequency variation ranges from 877 to 1009 Hz at the 10× dilution concentration, 631 to 743 Hz at the 50× dilution concentration, and 395 to 509 Hz at the 100× dilution concentration; while the frequency variation ranges from 50 to 54 Hz at the 10× dilution concentration, 38 to 41 Hz at the 50× dilution concentration, and 25 to 28 Hz at the 100× dilution concentration for negative samples, with almost no significant changes. The frequency variation of negative samples was 50~54 Hz at the 10× dilution concentration, 38~41 Hz at the 50× dilution concentration, and 25~28 Hz at the 100× dilution concentration, with almost no significant change, so the positive and negative samples could be clearly identified based on the frequency difference.

The host interface for performing the RNA virus detection test is shown in Figure 9. The biomolecule binding reaction and the duration of the assay can be observed in real time. The entire RNA virus detection process takes only 5 min from the time the sample is injected until the frequency curve levels off when the reaction reaches saturation. At the same time, the weak binding process of the DNA nanoprobe to the viral nucleic acid molecule can be transmitted into a significant change in frequency, indicating a high detection sensitivity.

## 4. Conclusions

In this study, the plasmonic resonance properties of the AuNRs dimer-quartz substrate system and the performance of the QCM biosensor chip were investigated numerically using finite element simulation software. The simulation results show that the arrangement and spacing of AuNRs dimers in the AuNRs dimer-quartz substrate system make the anisotropic gold nanorods exhibit extremely complex plasmonic resonance modes. It can be found that the elimination spectrum of gold nanorods in the parallel arrangement structure gradually red-shifts with the increase in the arrangement spacing D, and the resonance peak increases linearly. However, the extinction spectrum of gold nanorods in the longitudinally arranged structure gradually red-shifts with the decrease in D, and the growth rate of the resonance peak gradually increases. Based on this kind of AuNRs dimer-quartz substrate system, it can be applied to the design of biological nanosensors to detect biomolecules such as DNA and RNA. In addition, a biosensor chip without an electrode structure was designed in this study, and the simulation results verified the performance and feasibility of this structure. The electrodeless biosensor chip peels the metal electrode off the surface of the quartz wafer on the basis of the electrode, which reduces the influence of the quality of the metal electrode on the quartz wafer, widens the quality detection range of the electrodeless biosensor chip, and then improves the detection sensitivity.

The electrodeless biosensor chip adopts a ring-circular gold electrode structure, and then selects the appropriate arrangement and spacing of gold nanorods according to different detection needs, and assembles highly specific DNA primer probes on the surface of these gold nanorods. arranged on a quartz wafer, the electrodeless structure can transmit light in the middle area of the ring electrode, forming one transmission and two absorption phenomena of gold nanorods, thereby enhancing the optical signal intensity, improving the accuracy of measurement results, and realizing label-free detection. In a word, this study adopted the combination of plasma technology and electrodeless piezoelectric sensor chip with nanogram mass sensitivity.

## Figures and Tables

**Figure 1 micromachines-13-01523-f001:**
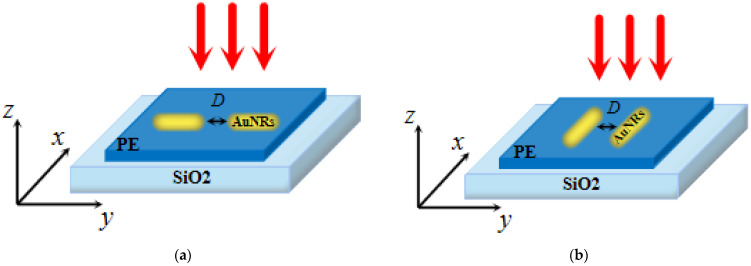
(**a**) Three-dimensional model of the longitudinally aligned gold nanorod dimer-PE-quartz structure; (**b**) three-dimensional model of the juxtaposed gold nanorod dimer-PE-quartz structure.

**Figure 2 micromachines-13-01523-f002:**
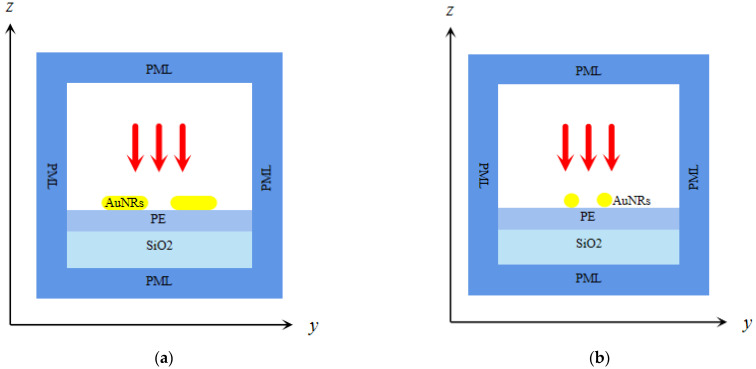
(**a**) The y-z section of the numerical simulation of the longitudinally arranged structure; (**b**) the y-z section of the numerical simulation of the side-by-side structure.

**Figure 3 micromachines-13-01523-f003:**
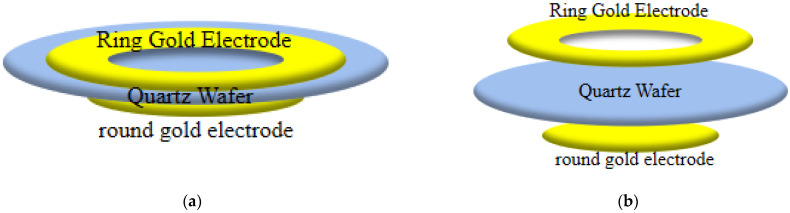
(**a**) Three-dimensional model of the QCM biosensor chip with a polar structure; (**b**) three-dimensional model of the QCM biosensor chip without an electrode structure.

**Figure 4 micromachines-13-01523-f004:**
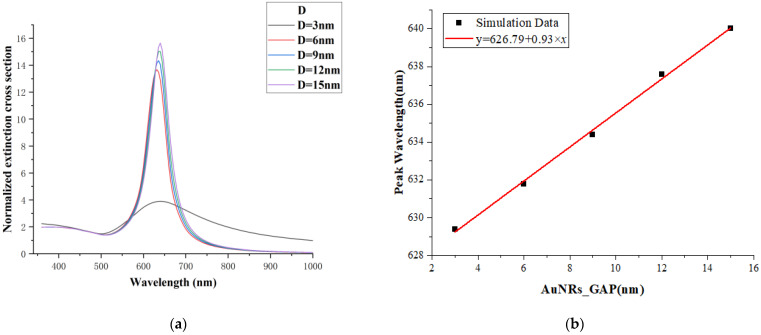
(**a**) The extinction cross-section of the gold nanorod dimer in the juxtaposed structure; (**b**) the relationship between the arrangement spacing of the gold nanorod dimer and the resonance peak wavelength.

**Figure 5 micromachines-13-01523-f005:**
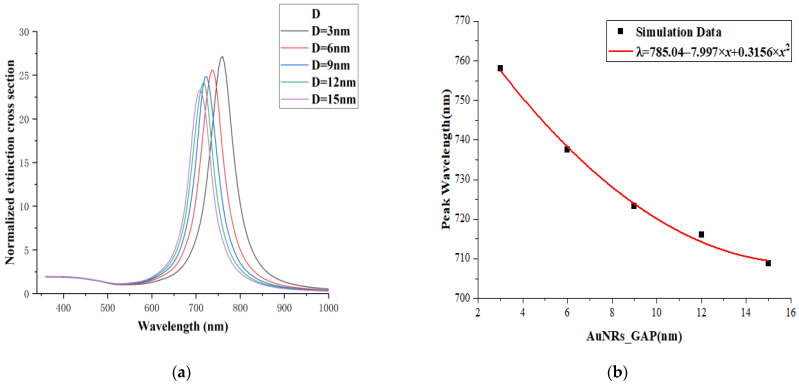
(**a**) The extinction cross-section of the gold nanorod dimer in the longitudinal structure; (**b**) the relationship between the arrangement spacing of the gold nanorod dimer and the resonance peak wavelength.

**Figure 6 micromachines-13-01523-f006:**
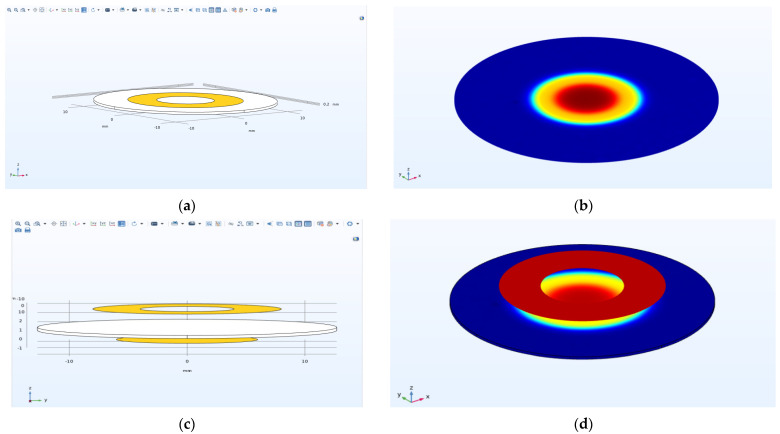
(**a**) Software modeling model of the polar biosensor chip; (**b**) surface potential distribution of the polar biosensing chip; (**c**) software modeling model of the electrodeless biosensing chip; (**d**) surface potential distribution of the electrodeless biosensing chip.

**Figure 7 micromachines-13-01523-f007:**
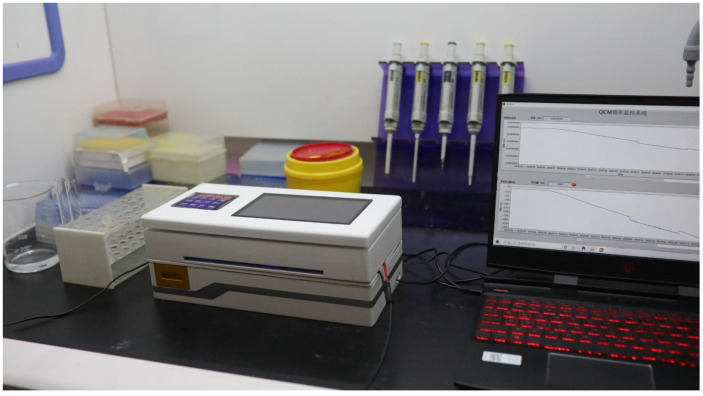
RNA virus experimental platform.

**Figure 8 micromachines-13-01523-f008:**
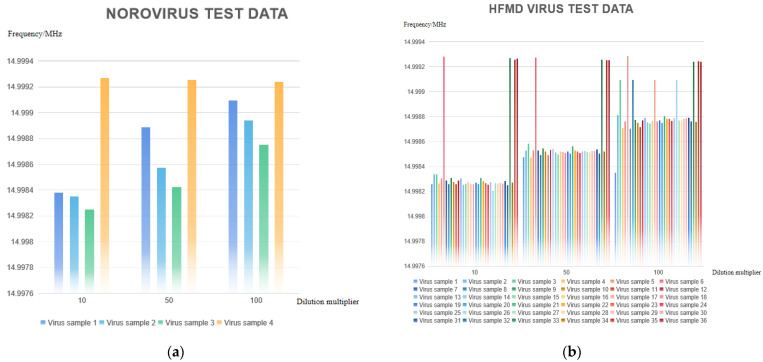
(**a**) Norovirus test results at different dilution concentrations; (**b**) HFMD test results at different dilution concentrations.

**Figure 9 micromachines-13-01523-f009:**
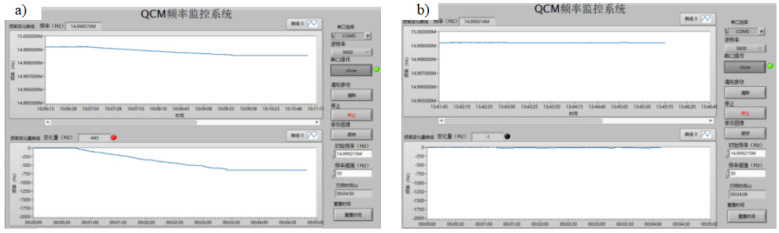
(**a**) Interface for testing RNA virus positive samples (The title of the diagram indicates: QCM frequency acquisition system.); (**b**) interface for testing negative RNA virus samples (The title of the diagram indicates: QCM frequency acquisition system.).

**Table 2 micromachines-13-01523-t002:** Electric field distribution under the resonance peak of the gold nanorod dimer in the juxtaposed structure (Y-X section).

AuNRs_GAP	3 nm	6 nm	9 nm	12 nm	15 nm
Electric FieldResonance Peak	629.4 nm	631.8 nm	634.4 nm	637.6 nm	640 nm
Ex		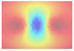		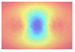	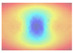
Ey	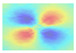	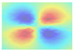	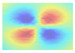	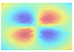	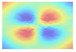
Ez	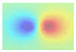	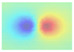	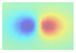	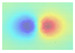	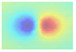
|E|	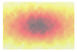				

**Table 3 micromachines-13-01523-t003:** Electric field distribution under the resonance peak of the gold nanorod dimer in the juxtaposed structure (X-Z section).

AuNRs_GAP	3 nm	6 nm	9 nm	12 nm	15 nm
Electric FieldResonance Peak	629.4 nm	631.8 nm	634.4 nm	637.6 nm	640 nm
Ex	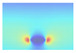	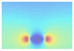	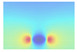	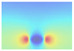	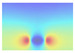
Ey	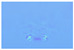	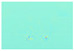	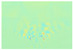	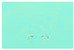	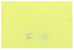
Ez	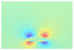	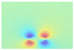	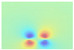	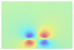	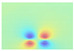
|E|	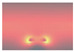	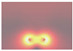	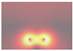	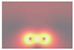	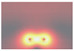

**Table 4 micromachines-13-01523-t004:** Electric field distribution under the resonance peak of the gold nanorod dimer in the longitudinal structure (Y-Z section).

AuNRs_GAP	3 nm	6 nm	9 nm	12 nm	15 nm
Electric FieldResonance Peak	758.1 nm	737.5 nm	723.2 nm	716.0 nm	708.8 nm
Ex	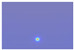	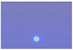	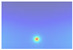	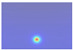	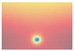
Ey	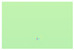	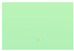	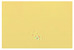	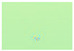	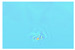
Ez	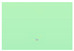	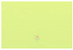	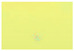	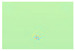	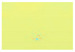
|E|	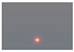	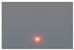	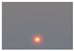	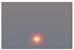	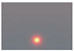

**Table 5 micromachines-13-01523-t005:** Electric field distribution under the resonance peak of the gold nanorod dimer in the longitudinal structure (Y-X section).

AuNRs_GAP	3 nm	6 nm	9 nm	12 nm	15 nm
Electric FieldResonance Peak	758.1 nm	737.5 nm	723.2 nm	716.0 nm	708.8 nm
Ex	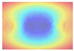	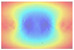	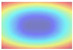	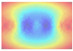	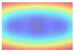
Ey	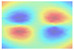	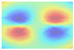	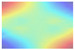	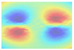	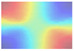
Ez	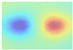	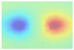	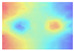	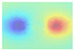	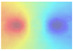
|E|	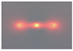	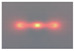	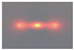	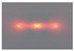	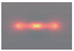

**Table 6 micromachines-13-01523-t006:** Electric field distribution under the resonance peak of the gold nanorod dimer in the longitudinal structure (X-Z section).

AuNRs_GAP	3 nm	6 nm	9 nm	12 nm	15 nm
Electric FieldResonance Peak	758.1 nm	737.5 nm	723.2 nm	716.0 nm	708.8 nm
Ex	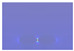	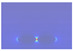	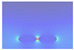	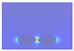	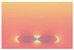
Ey		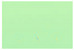	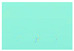	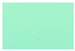	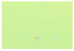
Ez	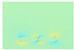	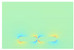	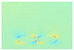	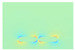	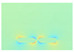
|E|	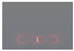	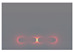	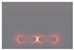	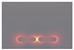	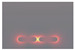

## Data Availability

Not applicable.

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
