# Peer review of "Research on Dual-Technology Fusion Biosensor Chip Based on RNA Virus Medical Detection"

_micromachines, 2022, doi:10.3390/mi13091523_

Round 1

Reviewer 1 Report

The paper „Research on Dual-technology Fusion Biosensor Chip based on RNA Virus Medical Detection” by Zhu and Xie describes design simulations of a QCM biosensor chip with gold nanorods (AuNRs). Experimental data are not included.

The paper covers an interesting subject, but some points remain to be discussed.

-       lines 23–25 “better sensing performance” and “to improve the sensitivity and accuracy of detection”: It remains unclear whether this refers to QCM or LSPR or the combination of both

-       The first paragraph of the introduction (lines 34–43) should be supported by references.

-       References in the text start with Ref. 3 (line 52), i.e., Ref. 1+2 are missing

-       Line 44: “Local surface plasmon resonance is a non-invasive optical detection technology”: This should be elaborated a bit more in context, since invasive/non-invasive is typically linked with the sample, not with the detection method

-       LSPR applications in RNA sensing are missing

-       LSPR and QCM parts in the introduction could be a little better coordinated in order to better highlight differences and similarities, including suitability to the respective application. For instance, lines 44–45 claim “real-time monitoring of the binding between biomolecules” for LSPR, while lines 80–81 lists “chemical adsorption reaction between the primer probe and the target biomolecule” for QCM, which gives the impression that different kinds of binding are observable with LSPR and QCM.

-       Regarding the combination of QCM with LSPR, the benefit remains unclear. Furthermore, the description of earlier works in this context is missing.

-       Lines 85–86: “The detection of ions provides an important solution with the potential to advance the practical application of QCM sensors [14-15]”. This statement remains unclear, and it is not supported by ref. 14-15 which focus on glucose detection.

-       Lines 94–104: The statements listed here should be supported by references.

-       Lines 200–201 / Section 3.2. / conclusion, biosensor chip with and without electrode: It remains unclear whether the quartz is supposed to act as QCM biosensor or just as substrate for the Au nanorods used for LSPR. In case of QCM biosensing: How is the QCM to be operated without electrode?

-       Can the improvements in sensitivity be quantified, e.g. by a factor? It partly remains unclear whether the improvement is compared to QCM, LSPR, or the combination of both

-       Lines 223-225: “Because virus samples need to be injected directly into the electrode of biochip, and different biological probes need to be replaced for different virus detection, which increases the chip consumption.”, Electrodeless QCM devices are responsive to electrical chances (e.g., ion strength, conductivity) in the surrounding. From an experimental point of view, this might be a problem, for instance, when switching between a carrier medium and a real sample or when injecting virus samples of different origin. It should be discussed how this issue would be handled in the “electrodeless biosensor chip”.

Reviewer 2 Report

This manuscript entitled “Research on Dual-technology Fusion Biosensor Chip based on RNA Virus Medical Detection” by Jin Zhu and Yushan Xie reports a computational study of localized surface plasmon resonance (LSPR) integrating with Quartz crystal microbalance (QCM). The authors simulated electric field distribution of different shapes of nanoparticle/nanorod on QCM substrate, resonance in different directions.    

As authors mentioned that “the equipment is complex and it is difficult to achieve synchronous measurement, which makes the results unable to achieve the true synergy and complementary effect” in line 108, it indicates nobody will combine SPR and QCM for biosensing at the same time. The authors may need to clarify why people need to know the interaction between these two technologies, and what kind of situation, the people will combine them together for sensing.

As the title named “Dual-technology Fusion Biosensor Chip”, I expected that I will learn how the acoustic wave interacts with surface plasmon resonance. However, in this study, I only read the QCM substrate effect to SPR signal, but without interactions between two physical parameters. I am so disappointed. I hope the authors can provide some study about that in this manuscript that will be valuable for readers.

Recently, several efforts about localized SPR biosensors from several groups, the authors may need to review them in this manuscript, such us:

(1)   https://doi.org/10.1016/j.bios.2020.112850

(2)   https://doi.org/10.1016/j.snb.2021.131327

(3)   https://doi.org/10.1016/j.snb.2022.132240

(4)   https://doi.org/10.1016/j.cej.2021.133864

(5)   https://doi.org/10.1016/j.bios.2021.113956

(6)   https://doi.org/10.1016/j.bios.2021.113672

Round 2

Reviewer 1 Report

The paper „Research on Dual-technology Fusion Biosensor Chip based on RNA Virus Medical Detection” by Zhu and Xie describes design simulations of a QCM biosensor chip with gold nanorods (AuNRs) to obtain additional LSPR characteristics. In the revised version, experimental data have been added, and the points to be discussed have been considered. In general, the review of the revised version would have been more convenient if parts of the newly added text had not only been cited in the cover/response letter. Instead, it should have been stated each time where in the revised version of the paper the newly added text had been added or – if no addition was made – why there was no addition.

A few points remain to be discussed; they are listed below (green font) in context with the original points (black font) and the authors’ responses from the cover/response letter (red font).

Point 1:lines 23–25 “better sensing performance” and “to improve the sensitivity and accuracy of detection”: It remains unclear whether this refers to QCM or LSPR or the combination of both

Response 1:This paper describes the combination of LSPR and QCM technologies for biosensing detection to improve the sensitivity and accuracy of RNA virus detection compared to a single LSPR biosensor or QCM biosensor.

ð  Reviewer’s comment to response: This information has not been included in the paper and, therefore, may remain unclear for the reader.

Point 3:Line 44: “Local surface plasmon resonance is a non-invasive optical detection technology”: This should be elaborated a bit more in context, since invasive/non-invasive is typically linked with the sample, not with the detection method

Response 3:A review of LSPR technology for virus detection is added in lines 71-86 of the article.

ð  Reviewer’s comment to response: OK, but the use of the term “non-invasive” in this context still remains unclear.

Point 9:Lines 200–201 / Section 3.2. / conclusion, biosensor chip with and without electrode: It remains unclear whether the quartz is supposed to act as QCM biosensor or just as substrate for the Au nanorods used for LSPR. In case of QCM biosensing: How is the QCM to be operated without electrode?

Response 9:The QCM chip is used as a biosensor in combination with gold nanorods with LSPR properties for RNA virus detection. The electrodeless structure in this paper is shown in Figure 3(b), which means that The electrodes are are separated from the upper and lower surfaces of the quartz wafer of the QCM and are formed on the inner surface of the measuring cell. The distance between the top electrode and the quartz wafer is 2mm, and the distance between the bottom electrode and the quartz wafer is 1mm. So it does not mean that there are no electrodes. For RNA virus detection, primer probe that binds specifically to the target viral nucleic acid molecule is fabricated on the surface of the quartz chip in an electrodeless structure. Compared with the traditional QCM structure with electrodes, the probe is directly prepared on the gold electrode. However, in the QCM without electrodes, the probe is prepared on the quartz wafer. When entering the school for RNA virus detection, it is only necessary to clean the quartz chip in the chip and reassemble the primer probe. It was verified through modeling simulations that after applying a certain voltage to produce the inverse piezoelectric effect, the high potential remains concentrated in the region of the QCM chip used to assemble the gold nanorods in the electrodeless structure, which provides the energy for the resonance of the QCM chip. Finally, the resonance frequency of the QCM chip with gold nanorods assembled with LSPR characteristics is used to detect the presence of the target virus in the sample solution to be tested. This section is added to the article in lines 251-260 and 400-405.

ð  Reviewer’s comment to response: OK, but please make sure that all of the information, which is required to understand the topic of the paper, is included in the paper.

Point 10:Can the improvements in sensitivity be quantified, e.g. by a factor? It partly remains unclear whether the improvement is compared to QCM, LSPR, or the combination of both

Response 10:Detection sensitivity, efficiency and accuracy is compared to single virus detection techniques. A related section on detection sensitivity is added to the article in section 3.3 - RNA virus detection tests.

ð  Reviewer’s comment to response:

o   Where exactly in the text can the comparison been found?

o   Please define HFMD virus (the term is in the text but not in context with the abbreviation)

o   Section 3.3.3 and Fig. 8: Please define more clearly (fig. caption or legend) what exactly the respective columns represent – is it a group of virus sample or a replication? If it is the mean: “each group was tested three times and then averaged” – where are the standard deviations? It remains unclear from Fig. 8, whether the positive virus samples can be distinguished from the negative samples.

o   How exactly does this experiment show the benefit of a QCM chip provided with gold nanorods to gain LSPR properties compared to separated QCM or LSPR detection for RNA virus detection?

Point 11:Lines 223-225208-210: “Because virus samples need to be injected directly into the electrode of biochip, and different biological probes need to be replaced for different virus detection, which increases the chip consumption.”, Electrodeless QCM devices are responsive to electrical chances (e.g., ion strength, conductivity) in the surrounding. From an experimental point of view, this might be a problem, for instance, when switching between a carrier medium and a real sample or when injecting virus samples of different origin. It should be discussed how this issue would be handled in the “electrodeless biosensor chip”.

Response 11:How to solve the problem of changing different primers and different nucleic acid samples, and how to react to different samples, the chip designed in this paper is processed as follows. For RNA virus detection, a primer probe that specifically binds to the nucleic acid molecule of the target virus was immobilized on the quartz chip surface of the electrodeless QCM. Since the gold electrode is made on the measuring cell, it does not contact the probe. When detecting different RNA viruses, it is only necessary to clean the quartz chip in the chip and reassemble the primer probe.

ð  Reviewer’s comment to response: It remains unclear how the problem of several influencing parameters on the signal response, i.e., mass loading and change in electrical parameters is solved, since they can cancel each other out regarding the resulting QCM signal response.

Reviewer 2 Report

It can be published in current form.

Author Response

Dear reviewer:
Thank you for your decision and constructive comments on my manuscript, I sincerely thank you and wish you all the best in your future work and life!
Sincerely.
Jin Zhu, Yushan Xie

Round 3

Reviewer 1 Report

OK